# Exploring the Use of Hydroxytyrosol and Some of Its Esters in Food-Grade Nanoemulsions: Establishing Connection between Structure and Efficiency

**DOI:** 10.3390/antiox12112002

**Published:** 2023-11-14

**Authors:** Josefa Freiría-Gándara, Tamara Martínez-Senra, Carlos Bravo-Díaz

**Affiliations:** Departamento Química-Física, Facultad de Química, Universidade de Vigo, 36310 Vigo, Spain; jfreiria@uvigo.es (J.F.-G.); tamartinez@alumnos.uvigo.es (T.M.-S.)

**Keywords:** hydroxytyrosol, food-grade nanoemulsions, lipid oxidation, antioxidant efficiency, partition constants

## Abstract

The efficiency of HT and that of some of its hydrophobic derivatives and their distribution and effective concentrations were investigated in fish oil-in-water nanoemulsions. For this purpose, we carried out two sets of independent, but complementary, kinetic experiments in the same intact fish nanoemulsions. In one of them, we monitored the progress of lipid oxidation in intact nanoemulsions by monitoring the formation of conjugated dienes with time. In the second set of experiments, we determined the distributions and effective concentrations of HT and its derivatives in the same intact nanoemulsions as those employed in the oxidation experiments. Results show that the antioxidant efficiency is consistent with the “cut-off” effect—the efficiency of HT derivatives increases upon increasing their hydrophobicity up to the octyl derivative after which a further increase in the hydrophobicity decreases their efficiency. Results indicate that the effective interfacial concentration is the main factor controlling the efficiency of the antioxidants and that such efficiency strongly depends on the surfactant concentration and on the oil-to-water (*o*/*w*) ratio employed to prepare the nanoemulsions.

## 1. Introduction

Use of natural ingredients to avoid food spoilage, particularly lipid oxidation, is an on-growing demand of consumers by total (or at least partial) substitution of synthetic additives [1,2,3,4]. Important sources of natural extracts are fruits and vegetables, and the food and pharmaceutical industry have focused on them because they are especially rich in bioantioxidants [4,5]. Olive oil is a natural, essential, component of the Mediterranean diet [6,7]. Huge quantities of antioxidant-rich olive by-products are generated during the harvesting of olive trees, and their incorporation into cosmetic and pharmaceutical formulations, and even in the food industry, is continuously being explored [7,8,9]. One of the most potent antioxidants (AOs) known in this field, which is the focus of the present work, hydroxytyrosol (HT), comes precisely from olive oil [10,11,12,13]. Its bioefficiency is a consequence of its chemical structure bearing a catechol (benzene-1,2-diol) moiety in which the H-atom of the carbon in the 4-position has been substituted by 2-hydroxyethyl group as shown in Figure 1.

In addition to its antioxidant properties, HT may also have antimicrobial effects and its intake by humans provides interesting benefits on health, including ani-inflammatory, antithrombotic effects, protecting human erythrocytes from hydrogen-peroxide-induced oxidative damage [14]. In addition, HT has been proven as an excellent candidate for the treatment of neurological diseases because of its ability to inhibit monoamine oxidase [13,15,16].

HT from olive leaves cannot be directly incorporated into foods, including parenteral emulsions, because of its characteristic flavor, which makes it palatably unaccepted, thus limiting its potential beneficial effects. But HT can be incorporated as an ingredient in formulations throughout its application in new systems of packaging or by encapsulation, and this opens the possibility of introducing chemically synthetized HT to exploit and take advantage of its health benefits and antioxidant properties [13,14].

As a matter of fact, in 2015 [17], the European Commission requested that the European Food and Safety Authority (EFSA) panel on Dietetic Products, Nutrition, and Allergies deliver an opinion about the potential use of chemically synthetized hydroxytyrosol and about its potential use as a novel food component that eventually could be added to foods, specifically to oils and margarines (question number EFSA-Q-2015-00749 raised by Seprox Biotech). The applicant intended to add chemically synthetized HT to oils and margarines even though the main dietary sources of HT are olive oil and table olives. The target group was the general population excluding children under 36 months of age and pregnant and breastfeeding women, and the use of oils and emulsions was intended to be at room temperature (with a warning for not heating oils) [17].

After some studies and discussions, and bearing in mind that HT is naturally present in olive oil and table olives as free HT, and in the conjugated forms of oleuropein and oleuropein-aglycones, the panel concluded that the use of HT as novel food is safe under the proposed uses [17]. Moreover, the panel resolved that the anticipated daily intake of HT is within the range of that associated with the consumption of olive oils and table olives, which has not been associated with adverse effects, concluding therefore that the consumption of chemically prepared HT is not nutritionally disadvantageous. Moreover, the antioxidant and antimicrobial character of HT, in addition to its bioactive potential, is leading the use of HT as a food ingredient: EFSA conveyed a positive opinion on the health claim related to potential health benefits and authorized the introduction of synthetic HT in the market as a novel food ingredient under the regulation EC-258/97 [17]. In summary, the present regulatory status of HT within the EU as an approved novel food ingredient opens the possibility of its use in various food and pharmaceutical formations in the near future [18,19,20]. Unfortunately, at present, the use of HT is almost restricted to the scientific level; however, promising results [14,21,22,23,24] may consolidate the addition of HT to oily-based products in the future to avoid lipid oxidation and other diseases.

HT is a relatively hydrophilic compound and its oral bioavailability could be strongly limited by an insufficient uptake from the mucosa due to its poor epithelial permeability [23,25,26]. Several strategies were proposed to incorporate it in biocompatible systems that improve its bioavailability profile in order to take full advantage of its health properties. Much attention has been paid to lipid-based formulations as microemulsions and nanoemulsions [27]. However, its bioavailability was not always optimal, showing problems for reaching its target system at sufficient concentrations [25]. For example, Mitsou et al. showed that the permeability efficiency of HT through an in vitro intestinal epithelium model constructed by Caco-2/TC7 and HT29-MTX cell lines decreases upon increasing the surfactant concentration of the emulsified system employed to encapsulate HT [25].

Other strategies to improve solubility of HT in non-aqueous systems (for example, in parenteral lipidic emulsions) and, eventually, its bioavailability, include the preparation of hydrophobic derivatives of HT by grafting inert alkyl chains through alkylation of the non-aromatic –OH group with carboxylic acids of different chain lengths [11,28,29]. This strategy, which has already been employed in the past to increase the hydrophobicity of different antioxidants, is precisely the focus of this work.

Lucas et al. [30] prepared hydrophobic derivatives of hydroxytyrosol and investigated their surface-active properties in oil–water mixtures in connection to the “cut-off” effect observed in emulsions. Although the systems were totally different, the surface tension results revealed that only hydroxytyrosol derivatives with an acyl chain length between C6 and C12 showed adequate surfactant properties, decreasing their *cmc* value upon increasing the alkyl chain length. Surprisingly, this non-linear dependency of surfactant properties observed with the increase in chain length of the HT esters fit quite well with their reported antioxidant efficiency by Medina et al. in *o*/*w* emulsions [31,32]. In addition, the antioxidant properties of a series of HT esters with different alkyl chain lengths (C4, C8, C12, and C18) were also investigated in liposomal systems. Particularly, the antioxidant efficiency of HT derivatives to protect liposomes from induced oxidation was influenced by the alkyl chain length, with the C12 ester derivative being the most efficient. Results suggested that the distribution of HT ester derivatives within the system could depend specifically on their chain length, and it could be related to different antioxidant behavior [33]. To provide insights into these puzzle-based observations—attributable to the differential incorporation of AOs in the different regions of multiphasic systems but without reporting their distributions—we synthesized a series of HT derivatives of different hydrophobicities while retaining their original catecholic structure, Figure 2, and investigated their antioxidant efficiency and their interfacial concentrations in nanoemulsions prepared with omega-3–enriched fish oil [22,34,35,36,37].

Furthermore, the benefits of biologically active omega-3 fatty acids, and, particularly, those of eicosapentaenoic (EPA) and docosahexaenoic (DHA) acids to human development, functioning, and health throughout our life-course, prompted us to investigate the behavior and antioxidant efficiency in nanoemulsions prepared with fish oils. The importance of EPA and DHA is so high for human health that a new index (EPA + DHA in red blood cells) was proposed to properly assess and evaluate their deficiency in human nutrition, detecting rapidly insufficient consumption which would eventually lead to deficiencies in diet, which are rather common in industrialized countries [22]. Martinez et al. [38] reported that a combination of HT with omega-3 fatty acids and curcumin improved pain and inflammation among breast cancer patients, and where HT plays a central role in the extent of the observed effects.

We have chosen nanoemulsions as an experimental platform in our studies because nanoemulsions constitute an advanced mode of drug delivery system that has been developed to overcome the major drawbacks associated with conventional drug delivery [18,39,40,41,42,43,44]. Because the food industry is now introducing omega-3 FA to prepare various kinds of functional foods in an attempt to provide health benefits over and above their basic nutritional aspects [45,46], there are considerable challenges in incorporating both antioxidants and omega-3 FAs into many types of functional food products due to their low water-solubility, poor chemical stability, and variable bioavailability [47]. Consequently, there has been growing interest in the development of appropriate delivery systems to encapsulate, protect, and release them [47,48].

Nanoemulsions offer a promising way to incorporate omega-3 fatty acids into liquid food systems like beverages, dressing, sauces, and dips. The composition and fabrication of nanoemulsions can be optimized to increase the chemical and physical stability of oil droplets, as well as to increase the bioavailability of omega-3 fatty acids. Delivery systems such as nanoemulsions could be used for a number of purposes: controlling lipid bioavailability; targeting the delivery of bioactive components within the gastrointestinal tract; and designing food matrices that delay lipid digestion and induce functional lipophilic constituents within the food and pharmaceutical industries. Nanoemulsion drug delivery systems can be, thus, envisaged as advanced modes for delivering and improving the bioavailability of hydrophobic drugs and drugs that have high first pass metabolism. The nanoemulsion can be prepared by both high energy and low energy methods, providing an excellent working system for optimal drug delivery for existing and newly developed antioxidants and antimicrobials, enhancing drug bioavailability, enabling site-specific drug targeting, and overcoming current limitations of drug formulations such as short elimination half-lives, poor drug solubility, and undesirable side effects [49,50]. Choosing nanoemulsions as a working platform has, at the same time, the advantage of attempting to fill the need for edible delivery systems to encapsulate, protect, and release bioactive components [39,40,48,51].

Our first aim here is, therefore, mainly focused on attempting to expand the use of HT and its derivatives in the fight against lipid oxidation and cognitive decline and cardiovascular diseases [52,53,54]. We are aware that this is only a first step—but necessary—to explore the use of HT derivatives in foods to prevent the oxidation of lipids and as biocide [55,56]. Research is also necessary to set the maximum dietary intake based on dietary surveys updated with composition data from scientific literature. Moreover, HT obtained from discarded by-products of the olive oil industry represents a valuable natural and cheap source of HT to prepare chemically modified derivatives, thus representing a real opportunity for adding further value to olive oil cultivars, leading to a much more sustainable olive oil production and minimizing ecological impact [52].

## 2. Materials and Methods

### 2.1. Materials

All chemicals were used as collected and of the maximum purity available. Deionized water (conductivity < 0.1 mS cm^−1^) was used in all experiments. Aqueous solutions were buffered with citric acid/citrate (0.04 M. pH∼3.0) to control acidity and to remove traces of metals that, eventually, could be present in the oil.

The fish oil (Omegatex 3020; 300 mg/g EPA and 200 mg/g DHA), was a generous gift of Solutex (Zaragoza, Spain). It was stripped of its natural tocopherols by passing it three times through a previously activated Al_2_O_3_ column. Total removal of tocopherols was confirmed by HPLC (IUPAC method 2.432). The stripped oil was kept in the dark, at low temperature, and in an inert atmosphere until its use in the preparation of the nanoemulsions. Further details are available [57,58].

4-*n*-hexadecylbenzenediazonium tetrafluoroborate, 16-ArN_2_BF_4_ was synthetized from commercial 4-*n*-hexadecylaniline (Aldrich 97%), and its purity was confirmed by NMR [59]. The commercially available radical 2,2 Diphenyl-1-picrylhydrazyl (DPPH^●^), was from Aldrich. Hydroxytyrosol (HT), and the esters and fatty acids employed in the preparation of hydrophobic HT esters, Figure 1, were acquired from Sigma-Aldrich or Across Organics and used as received.

Figure 2 illustrates the enzymatic routes employed for the synthesis of the antioxidants according to published procedures at T = 35 °C [11,58]. The Novozym 435 candida antarctica lipase employed in the synthesis of the HT derivatives was a generous gift of Novozymes (Bagsvaerd, Denmark) and used without further manipulation. A direct esterification procedure of HT with the corresponding fatty acid was employed to prepare the C_1_–C_6_ derivatives; meanwhile a transesterification procedure employing the corresponding fatty esters was employed to synthetize the C_8_–C_16_ derivatives. 

In a typical experiment, HT (3.24 mM) was added to a mixture of fatty acid or fatty ester (9.72 mM) and 33 mL benzene containing 0.972 g of Novozym 435 in a dry round bottom flask and stirred for 2–4 days. The enzyme was removed by decanting off the solution, 80 mL of diethyl ether was added, and the combined solution was extracted with aqueous 0.6 M Na_2_CO_3_ (3 × 30 mL) and dried over Na_2_SO_4_. The solvent was evaporated and the product was purified by flash column chromatography over silica gel using ethyl acetate/petroleum ether as the eluent (1:1 (v:v) for the C_1_–C_6_ and 2:1 (v:v) for the C_8_–C_16_ HT derivatives). ^1^H and ^13^C NMR spectra for HT and its alkyl esters (see Appendix A) agree with literature results [11,60,61]. Final yields were higher than 50%.

### 2.2. Methods

#### 2.2.1. Nanoemulsion Preparation

Nanoemulsions of different oil-to-water ratios (1:9, 2:8, 3:7, 4:6, and 5:5, v:v, V_T_ = 10 mL) were prepared by employing acidic water (0.04 M citrate buffer, pH 3.0), stripped fish oil, and the emulsifier Tween 80 (0.5–4% *w*/*w*) following the protocols described in detail elsewhere [62]. Briefly, hydrophilic AOs were dissolved in the aqueous buffer phase; meanwhile, the hydrophobic ones (≥C_4_) were dissolved in the oil phase. The solutions were mixed at high speed (20,000 rpm for 1 min) at room temperature and passed three times through a high-pressure homogenizer (Microfluidics, LM20 Microfluidizer) operating at 25,000 Psi, and the resulting nanoemulsions were translucid. The average droplet size was estimated to be ∼100 ± 20 nm, as determined by light scattering, and their stability was checked visually for at least 24 h, a much longer time than that required to complete the kinetic experiments (see below).

#### 2.2.2. Physical Characterization of Nanoemulsions: Droplet Size, Polydispersity Index, and ζ-Potential

The droplet size and polydispersity (PDI) of the different nanoemulsions were by employing dynamic light scattering (DLS, Malvern Mastersizer MS 3000 (Malvern Instruments Ltd., Worcestershire, UK)) at T = 25 °C. Prior to analyses, each nanoemulsion was diluted with citric-citrate buffer solution and the obtained solution was transferred immediately to a cuvette for the droplet size measurement. To determine the droplet size distribution, the refraction index values of 1.48 and 1.33 were used for fish oil and water, respectively. Measurements were performed in triplicate and reported values were given as D3,2 and as average ± standard deviation.

The ζ-potential was determined by employing a particle electrophoresis instrument (Zetasizer Nanoseries Nano-ZS, Malvern Instruments, Worcestershire, UK). Freshly prepared emulsions were diluted to a droplet concentration of 0.001% (*w*/*v*) with the citric-citrate buffer solution employed in the preparation of the emulsion. Results are reported as average ± standard deviation.

#### 2.2.3. Effects of Hydrophobicity on EC_50_ Values

Previously reported results by us and others [11,60,61] confirmed that grafting HT and other antioxidants with inert alkyl chains did not result in significant changes in their antioxidant activity against the commercially available DPPH radical. To check if the same insignificant changes hold for HT derivatives, we further tested the point by determining the EC_50_ values for the different HT derivatives. A UV-Vis Powerwave XS Microplate Reader (96-well microplate, Bio-Tek Instruments, Inc., Winooski, VT, USA) was employed to monitor the loss of DPPH^●^ (from reaction with each AO) at T = 25.0 ± 0.1 °C. The absorbance of each well was recorded (λ = 515 nm) at 60 s intervals for a 4000 s period, Figure 3.

Each set of experiments was performed in triplicate and, in some instances, in quadruplicate, and the reported EC_50_ values in Table 1 (determined from data in Figure 3) are the average of these repetitions. Results confirm that alkylation of HT has a negligible effect on EC_50_ values, i.e., it does not affect the reactivity against DPPH^●^.

#### 2.2.4. Antioxidant Efficiency: Relative Oxidative Stability of Fish Oil Nanoemulsions

Accelerated tests are commonly used to assess the relative stability of emulsions because of their reproducibility [63]. In the present work, we used the AOCS official standard method Ti-1a-64 to determine the relative antioxidant efficiency of the antioxidants under a set of standardized conditions [64]. For this purpose, we measured the formation of conjugated dienes (CD) with time following the procedure described in detail elsewhere [63]. Previous experiments showed that the prepared nanoemulsions were stable for several weeks. We, however, controlled their stability visually; no phase separation was detected.

#### 2.2.5. Antioxidant Distributions in Binary Oil–Water Systems: Partition Constant *P*_W_^O^ in the Absence of Emulsifier

The partition constants, *P*_W_^O^, of HT and some of its esters were determined, by employing the shake-flask method [65], in binary stripped fish oil–water mixtures in the absence of emulsifier. Sets of four 1:9 (v:v, oil:water, total volume = 10 mL) binary mixtures, each one containing the same AO, were prepared by stirring the mixture at high speed for 1 min. HT and the C1–C3 derivatives were dissolved in the buffered aqueous solution; meanwhile, those with longer alkyl chains were dissolved in the fish oil phase. The final stoichiometric concentration of AOs was 3.5 mM in all samples. Mixtures were stirred at high speed for 2 min and then equilibrated at T = (25 ± 1) °C for at least 24 h in the dark. The resulting two phases were then divided by centrifugation. Concentrations of the AOs in each phase were estimated by interpolation in previously prepared calibration curves by employing UV-Vis (*λ* = 285 nm). Further details on the method have been published elsewhere [66].

The reported *P*_W_^O^ values in Table 2 are average of the four runs. The *P*_W_^O^ values for the more hydrophobic antioxidants are not reported because of the high uncertainty in their values, as the ratio of percentages may be susceptible to small errors in the determination of the values of the AO concentrations in the aqueous phase. In any case, *P*_W_^O^ values higher than 150 indicate that AOs are, essentially, water insoluble.

#### 2.2.6. Determining Distribution of HT and Its Derivatives in Intact Fish Oil Nanoemulsions

The distribution of HT and its derivatives was determined by using a chemical kinetic method, as have been published elsewhere [66,67] based on the reaction between a chemical probe (16-ArN_2_^+^, specifically localized at the interface) and the AOs. The changes detected in the observed rate constant for this reaction (*k*_obs_) at different surfactant volume fractions (Φ_I_) were evaluated according to the pseudophase kinetic model developed by Romsted and Bravo-Díaz and explained in detail elsewhere [66,67].

The chemical probe 16-ArN_2_^+^ reacts, virtually, with any antioxidant as described elsewhere and the reaction can be monitored by UV-Vis with the aid of a specially developed derivatization protocol [66]. Briefly, we employed the coupling agent *N*-(1-naphthyl)ethylenediamine dihydrochloride, NED, to quench the 16-ArN_2_^+^ ions that did not react with the antioxidants. The absorbance of the resulting azo dye was monitored at *λ* = 572 nm once diluted with a 50:50 (v:v) BuOH:EtOH mixture. This dilution step is critical and its purpose is to obtain an optically transparent, homogeneous solution. Details of the complete procedure have been published elsewhere [66].

Succinctly, after initiating the reaction between 16-ArN_2_^+^ and the antioxidants in the intact emulsion, constant volumes (200 µL) of the nanoemulsion were extracted at given time periods and rapidly added to numbered glass tubes to quench the reaction. Because the quenching reaction is much faster than the reaction between 16-ArN_2_^+^ and the antioxidants, the absorbance of the formed azo dye is proportional to the concentration of unreacted 16-ArN_2_^+^ so that the decrease in absorbance of the azo dye with time can be used to determine *k*_obs_. For this purpose, we fitted the experimental (absorbance, time) pairs of data to the integrated first-order rate equation by using a non-linear least squares method provided by a commercial computer program. Figure 4 illustrates the typical variations of the absorbance of the azo dye versus time as determined in the intact fish oil-in-water nanoemulsions. The solid lines are the fitting curves to the integrated and linearized first-order equations. The great fits of the (A, t) data pairs to the first-order kinetic equation demonstrate that the effectiveness of the quenching reaction in halting that with the antioxidant and that changes in the droplet sizes, if any, are not kinetically significant because otherwise random values should be obtained [66].

The relationships between *k*_obs_ and the partition constants of the AOs between the oil-interfacial, *P*_O_^I^, and aqueous-interfacial, *P*_W_^I^, regions are given by Equations (1)–(3), and the partition constants were estimated by fitting the variations of 1/*k*_obs_, with Φ_I_ at constant stoichiometric concentration of the antioxidant ([AO_T_] = 2.5 × 10^−3^ M), Φ_W_ (Φ_W_ = 0.9) and Φ_O_ (Φ_O_ = 0.1) as explained in detail elsewhere [66,67].


(1)
$$k_{\mathrm{obs}}=\frac{k_{I}{\left[\mathrm{AO}\right]}_{T}P_{W}^{I}}{\Phi_{I}P_{W}^{I}+\Phi_{W}}$$



(2)
$$k_{\mathrm{obs}}=\frac{k_{I}{\left[\mathrm{AO}\right]}_{T}P_{O}^{I}}{\Phi_{I}P_{O}^{I}{+\Phi }_{O}}$$



(3)
$$k_{\mathrm{obs}}=\frac{\left[{\mathrm{AO}}_{T}\right]k_{I}P_{W}^{I}P_{O}^{I}}{\Phi_{O}P_{W}^{I}+\Phi_{{}_{I}}P_{W}^{I}P_{O}^{I}+\Phi_{W}P_{O}^{I}}$$


#### 2.2.7. Statistical Analysis

All kinetic experiments were run in duplicate or triplicate for at least 2–3 t_1/2_. The determined *k*_obs_ values were <9%, with typical correlation coefficients of >0.997. All oxidation kinetic experiments were run in triplicate. Statistical analysis was carried out by one-way analysis of the variance (ANOVA, with Tukey’s HSD multiple comparison), setting the level of significance at *p* < 0.05. For this purpose, the SPSS 21.0 software was employed. Data in tables are given as (mean values ± standard deviation).

## 3. Results

### 3.1. Droplet Size and Polydispersity of the Prepared Emulsified Nanoemulsions and Their Physical Stability

DLS plot for the coarse fish oil emulsions (Figure 5A) showed that they were highly polydisperse, with average droplet size (D3,2) of d = 300 ± 20 nm at Φ_I_ = 0.038. The average droplet size was reduced significantly in the prepared fish nanoemulsions by employing the microfluidizer, showing a monomodal droplet size distribution. Figure 5B shows that the mean droplet size of fish nanoemulsions decreases upon increasing the emulsifier volume fraction. For instance, the average droplet size is d = 220 nm at Φ_I_ = 0.005, meanwhile d = 85 nm at Φ_I_ = 0.038. Hydroxytyrosol esters did not significantly affect the mean droplet size values of the prepared nanoemulsion when compared with the control (in the absence of antioxidants).

The polydispersity index (PDI) values are lower than 0.2 for all prepared nanoemulsions, Figure 5C, and they can be considered as monodisperse.

ζ-potential values of the prepared nanoemulsions are low, as otherwise expected for emulsions stabilized with non-ionic surfactants like Tween 80, with values ranging from −12 to −17 mV and remain essentially constant upon increasing the surfactant concentration, Figure 5D.

During storage at T = 4 °C in the dark, there was no phase separation for at least 60 days, which was a much longer time than that required to complete the kinetic experiments (see below). Figure 6A shows the variation of the mean droplet size with time as a function of the emulsifier volume fraction. Droplet size increased slightly with time and reached a constant value after 3–4 days, with values ranging 100–300 nm depending on the emulsifier volume fraction. PDI values increased slightly with time, reaching a constant value after 3–4 days, but were lower than 0.3, suggesting that storage did not significantly affect the distribution of droplet size (Figure 6B). Overall, results show that nanoemulsions are rather stable with ageing for more than 2 weeks.

### 3.2. Determining Partition Constants in Intact Nanoemulsions

Figure 7 shows the variations in *k*_obs_ with Φ_I_ for the reaction between 16-ArN_2_^+^ and the antioxidants in intact 1:9 oil:water fish oil nanoemulsions. The experimental (*k*_obs_, Φ_I_) pairs of data were fitted to the equations derived from the pseudophase model Equations (1)–(3), which hold for very hydrophilic (Equation (1)) and very hydrophobic (Equation (2)) antioxidants and for those of intermediate hydrophobicity (3).

The superb fits of both (*k*_obs_, Φ_I_) and (1/*k*_obs_, Φ_I_) data sets show that the hypotheses underlying the use of pseudophase models are fulfilled. As shown in Figure 7A–D, *k*_obs_ values decrease asymptotically with Φ_I_; meanwhile, the 1/*k*_obs_ values rise linearly with Φ_I_. The decreases in *k*_obs_ upon increasing Φ_I_ (6–10 fold) are similar to those found for other antioxidants and emulsions [65,67,68]. The values of the partition constants *P*_O_^I^ and *P*_W_^I^ and of the interfacial rate constant *k*_I_ are collected in Table 3.

It is worth noting that the *k*_I_ values, which represent the medium contribution to the overall rate of the reaction, are, essentially, independent of the length of the alkyl chain of the antioxidant, with an average value of *k*_I_ = (25 ± 2) × 10^−2^ M^−1^ s^−1^, strongly suggesting that the catecholic moiety of the antioxidants is sampling in a region with similar solvent characteristics (i.e., a similar location within the interfacial region).

Values of *k*_I_ in Table 3 reflect the high sensitivity of the chemical probe towards the chemical nature of the AOs [59,66]. *k*_I_ values for the C3-C16 HT derivatives are very similar because they all have the same catecholic moiety. At this very moment, we do not have a plausible explanation for the lower value of *k*_I_ for HT. Such a difference has been observed previously in other oils and antioxidants, and has been tentatively attributed to the changes in the chemical structure between HT and the alkyl derivatives. However, the average *k*_I_ value is quite different from that obtained in the presence of other antioxidants in similar emulsified systems; for example, it is ∼23 times higher than the average *k*_I_ value found for the reaction with caffeic derivatives (*k*_I_ = 1.1 × 10^−2^ M^−1^ s^−1^), but similar to that for the reaction with gallic acidderivatives, *k*_I_∼20 × 10^−2^ M^−1^ s^−1^ [65].

*P*_W_^I^ and *P*_O_^I^ values in Table 3 range ∼13–229, indicating that the removal of AOs from the oil and from the aqueous regions to the interfacial one is spontaneous (ΔG_transfer_ < 0). Thus, our results highlight the likely tendency of the HT esters to distribute preferentially in the interfacial region, as found for many other AOs [66]. *P*_W_^I^ values increase with increasing AO hydrophobicity, in keeping with the expected decrease in their aqueous solubility. However, *P*_O_^I^ values decrease upon increasing the hydrophobicity of the AOs, suggesting that, upon lipophilization, antioxidants are better solubilized in the oil region than in the interfacial region when compared with the less hydrophobic ones. According to the group-contribution theory, each methyl makes a constant contribution to the hydrophobicity (reflected as log *P*_O_^I^) of the antioxidants, Equation (4), and where the parameter A stands for the contribution of the non-alkyl part of the molecule to *P*_O_^I^, B the contribution of the methylene group; and n_CH2_ stands for the number of methylene groups in the alkyl chain. Figure 8 illustrates a typical Collander plot [69,70] showing the linear relationship according to Equation (4).



(4)
$$\log P_{O}^{I}={A+\mathrm{Bn}}_{{\mathrm{CH}}_{2}}$$


### 3.3. Distribution and Effective Concentrations of the Antioxidants in Fish Oil Nanoemulsions

The values of the partition constants *P*_O_^I^ and *P*_W_^I^ in Table 3 were used to estimate the distribution of the AOs in the intact nanoemulsions. Details of the calculations and the equations employed are not given here to keep focus and for the sake of clarity, but can be easily found elsewhere [66,67].

Results in Figure 9 support two important findings. (1) Increasing the emulsifier volume fraction makes the percentage of AOs in the interfacial region (%AO_I_) increase so that at Φ_I_ = 0.04, >70% of all AOs are located in that region. This result is quite relevant because the volume of the interfacial region represents, at most, only 4% of the total volume of the nanoemulsions. (2) The hydrophobicity of the HT esters has, at any Φ_I_ value, a large effect on the percentages of AOs in the interfacial region as shown in Figure 9B, following the apparently peculiar order: %HT_I_ < %C3_I_ < %C16_I_ < %C12_I_ < %C10_I_ < %C8_I_. Results clearly show, as demonstrated in previous papers [65,66], that %AO_I_ does not correlate directly with the hydrophobicity of the AOs but with its effective interfacial concentration [67].

Figure 10 shows the variations in the interfacial molarities, calculated by using Equation (5), as a function of the surfactant volume fraction. In Equation (5), n_T_ represents the total number of moles of the antioxidant, V_T_ the total volume of the emulsion, and Φ_I_ is the interfacial volume fraction. To distinguish between the stoichiometric and effective concentrations, we employed square brackets, [ ], defined as molarity of total emulsion volume, and parenthesis, ( ), defined as molarity of the interfacial region, respectively. The reader must be aware that V_I_, is the sum of *all* of the interfacial volumes of all of the aggregates present in the system, and that it matches that of the total volume of the added surfactant [66]. Emulsion droplets and other association colloids as micelles present in the system are broken and reformed at rates much faster than those of the relevant chemical reactions, allowing rapid transfer of materials between them. That is, they are in dynamic equilibrium [67,71,72,73].
(5)$$\left({\mathrm{AO}}_{I}\right)=\frac{n_{I}}{V_{I}}=\frac{n_{T}{(\%\mathrm{AO}}_{I})}{V_{I}}=\frac{\frac{n_{T}}{V_{T}}{(\%\mathrm{AO}}_{I})}{\frac{V_{I}}{V_{T}}}=\frac{{[\mathrm{AO}}_{T}{](\%\mathrm{AO}}_{I})}{\Phi_{I}}$$

Results in Figure 10 lead us to conclude that the effective interfacial concentration, (AO_I_), of any HT ester is 50–250 times higher than the stoichiometric concentrations of the AOs. That is, the interfacial region concentrates the AOs because of its small volume. (AO_I_) values depend strongly on the surfactant volume fraction, Φ_I_, but also on the hydrophobicity of the AO and on the *o*/*w* ratio employed in the preparation of the nanoemulsions. These are, indeed, remarkable outcomes because of their potential consequences on the rate of the inhibition reaction between the AOs and the radicals, which depends on the chemical structure of the antioxidant (computed in the rate constant) and on the interfacial concentrations. It is worth noting that, as we have shown in earlier reports, the rate of the inhibition reaction does not depend on droplet surfaces or sizes [65,66,68].

The reader must be aware that, even if the %AO_I_ value is low, the effective interfacial concentration may be much higher than the effective concentrations in the aqueous and oil regions, and certainly much more than the stoichiometric molarity. This is, in part, why antioxidants are so effective and how we can improve their efficiency while keeping the stoichiometric concentration constant.

Figure 11 shows the variation in the effective interfacial concentration of the antioxidants with the length of the alkyl chain at three surfactant concentrations (constant *o*/*w*, Figure 10A) and at variable *o*/*w* concentrations (constant Φ_I_ = 0.005, Figure 11B). At a constant *o*/*w* ratio, the highest increase in the interfacial concentration is achieved at the lowest surfactant fraction (Φ_I_ = 0.005, Figure 10A), resembling its variation with the alkyl chain length that of a typical cut-off profile, with an effective interfacial concentration for the C8 derivative about 100 times higher than the stoichiometric concentration. It is noticeable, however, that the interfacial concentration remains essentially constant with the alkyl chain length when employing high surfactant concentrations. The reason of this behavior lies in the fact that at low Φ_I_ values (i.e., low interfacial volume), the fraction of surfactants incorporated into the interfacial region is not very high, see Figure 8, and the percentage of incorporation of AOs depends critically on the length of the alkyl chain. Meanwhile, at high Φ_I_ values, antioxidants are totally incorporated into the interfacial region no matter the length of the alkyl chain, and their effective concentrations are much lower (∼20 times higher than the stoichiometric region) because of the much higher interfacial volume.

Figure 11B shows the critical and rather complex role of the oil-to-water ratio on the effective interfacial concentrations. When emulsions are prepared with a ratio 1:9 (o:w), the typical cut-off effect is observed; however, the effect is diluted upon increasing the oil proportion and it is not noticeable when equal amounts of oil and water are employed. Results, thus, suggest that the effective interfacial concentrations can be modulated by changing the oil-to-water ratio.

### 3.4. Relative Oxidative Stability of Fish Oil Nanoemulsions in the Presence of HT Derivatives

We estimated the relative oxidative stability of fish oil nanoemulsions by controlling the extent of lipid oxidation under a set of standardized conditions. For this purpose, we measured the level of oxidation on the basis of the production of conjugated dienes (CD) with time. The method, identical to the AOCS official standard method Ti-1a-64, is rather sensitive and reproducible and suitable for monitoring the early stages of lipid oxidation [63]. Figure 12A is illustrative of a typical oxidation kinetics experiment and where one can observe a constant small increase (lag time) up to a sudden increase in the formation of conjugated dienes (CD). The relative efficiency was assessed by determining the necessary time to reach an increase in CD production of 0.5%. Further details on the experimental procedure and calculations can be found elsewhere.

Figure 12B shows the variation in the production of CD (increase of 0.5%) for the different HT esters. The efficiency order found in fish nanoemulsions is C8 > C10 > C16 > C3 > HT >> control (in the absence of AO). This apparently anomalous behavior has already been described in the literature and illustrates the so-called “cut-off” effect, indicating an increase in the efficiency of antioxidants upon increasing their hydrophobicity up to a maximum (in this case, C8) after which the efficiency decreases upon increasing the length of the alky chain (i.e., the hydrophobicity of the AO). Results also show that the antioxidant efficiency of AOs increases by decreasing Φ_I_ in line with the observed effect of Φ_I_ on the interfacial antioxidant concentration (AO_I_), Figure 11A. Note that (AO_I_) values depend on both the percentages of AOs in the interfacial region %AO_I_ and Φ_I_, Equation (5), and both parameters have an opposite effect and the increase in the emulsifier volume fraction is always higher than the increase in the percentage of AOs in the interfacial region; for example, an increase in Φ_I_ from 0.005 to 0.4 would increase the interfacial volume ∼8 times, but an increase in the percentage of AO in the interfacial region would only increase it ∼1.8 times and, therefore, AOs are diluted.

### 3.5. Structure–Efficiency Relationships

Bringing together the results of both sets of kinetic experiments (Figure 13), one can analyze the causes of the variable efficiency of the antioxidants. This is, certainly, related to the hydrophobicity of the antioxidants, as it is the only experimental difference between the various kinetic runs. Such a change in the hydrophobicity does not convey a change in reactivity because the moiety of all antioxidants is the same (as expected from the DPPH experiments, Section 2.2.3) and because the reactive moiety of the AOs is sampling in the same region of the nanoemulsions as shown by the constancy in the *k*_I_ values (Section 3.2). However, changing the hydrophobicity of the antioxidants makes them distribute in a different way, modifying their local concentrations at the oil, interfacial, and aqueous regions. As noted before, the concentration of AOs in the aqueous region is low and one should not expect peroxyl radicals to be located in such a region. Hence, the contribution of the rate of the inhibition reaction in the aqueous region should be negligible. In previous works, we showed that that the reaction in the oil region, which could be plausible as the concentration of both antioxidants and peroxyl radicals may not be negligible, is not important. The reason is that peroxyl radicals, which have a dipole moment much higher than the parent lipid, diffuse towards more polar regions (i.e., the interfacial region). Thus, there should be a direct relationship between their efficiency and their local concentrations. Figure 13 is illustrative and shows that the variations of the interfacial concentration (AO_I_), the percentage of AOs in the interfacial region, %AO_I_, and the relative efficiency of the antioxidants, as assessed by the extent of the lag time, parallel each other, with a maximum at the C8, at two surfactant volume fractions.

## 4. Conclusions

Overall results show that HT and its hydrophobic esters are excellent candidates as potential food additives because their efficiency in inhibiting lipid oxidation is very high and improves significantly upon lyophilization. The octyl derivative is the most efficient, with an increase in the oxidative stability of nanoemulsion of ∼10 times when compared with the control (Φ_I_ = 0.005). Results also show that the influence of the alkyl chain length in the antioxidant efficiency of HT and its ester derivatives obtained here is fully consistent with their interfacial concentration obtained. Both parameters—interfacial concentrations and antioxidant efficiency—are parallel, demonstrating that there exists a direct connection among them. Separately, the direct relationship of antioxidant efficiencies and their distributions reveals the importance of the surfactant volume fraction, the main parameter being controlling the antioxidant efficiency. For example, for C8 ester derivative, an increase in Φ_I_ of ∼8 times leads to a decrease in its efficiency of ∼1.5 times because its interfacial concentrations decrease. Changes in their interfacial concentrations of HT or very hydrophobic AOs (i.e., C12, C16) have also been observed by modifying the *o*/*w* ratio of the emulsions but not those of intermediate hydrophobicity (C3). A decrease in the *o*/*w* ratio enhances the interfacial concentration of hydrophobic ester derivatives, but it decreases the hydrophilic ones. Results demonstrate that the distribution of AOs in a compartmentalized system should be taken into consideration to predict their potential antioxidant efficiency.

To sum up, the introduction of inert lipophilic chains in HT while retaining the catecholic moiety opens the possibility of using HT derivatives in a variety of formulations, from parenteral emulsions to creams to preparations to fight against cancer. Making HT derivatives hydrophobic also improves their bioavailability, making the results obtained in this work highly important for an optimum design of omega-3–enriched emulsions, which are very susceptible to suffering radical oxidation and the generation of potentially harmful products. Moreover, since HT is now an authorized novel food ingredient (in the European Union) that can be added to different food-stuffs, results also open new possibilities for revalorizing waste products from oil production because both olive oil and olive leaves are rich HT sources.

## Figures and Tables

**Figure 1 antioxidants-12-02002-f001:**
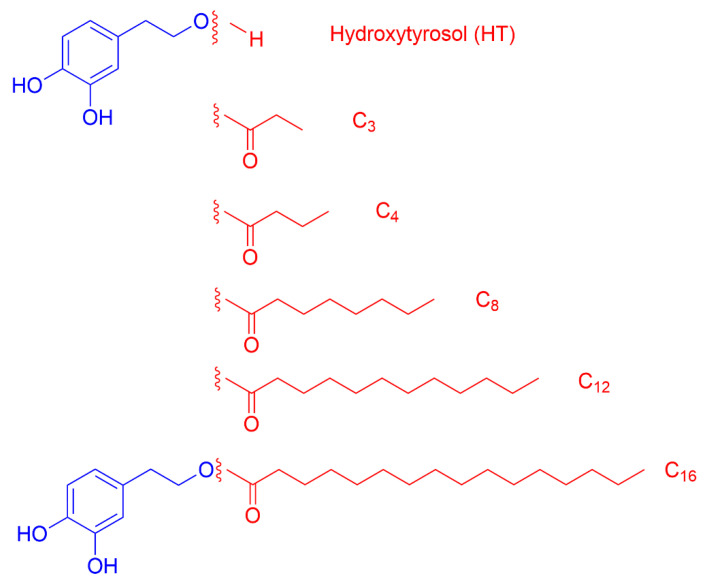
Hydroxytyrosol derivatives employed in this work.

**Figure 2 antioxidants-12-02002-f002:**
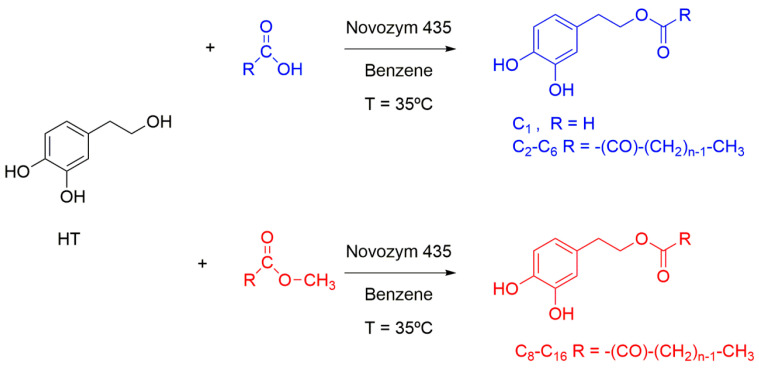
Synthetic routes used to prepare HT esters of different hydrophobicities.

**Figure 3 antioxidants-12-02002-f003:**
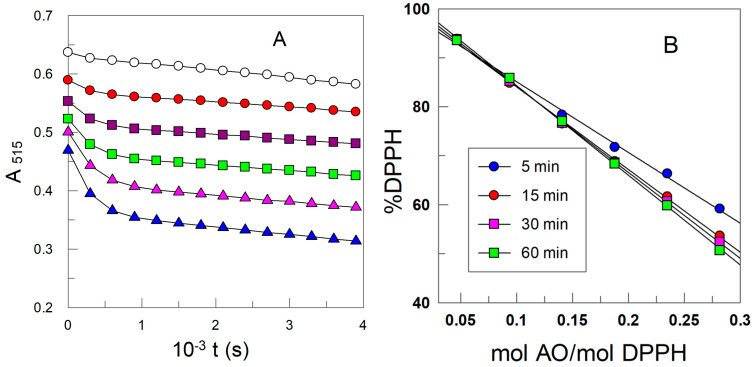
(**A**) Changes in the absorbance of DPPH^●^ with time at six C3/DPPH^●^ mole ratios-: 
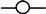
 0.047, 
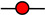
 0.094, 
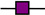
 0.141, 
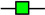
 0.188, 
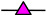
 0.235, 
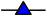
 0.282, T = 25 °C. Solid lines were drawn to aid the eye. (**B**) Variation in the percentage of DPPH^●^ as a function of the AO/DPPH^●^ ratio at different reaction times and the corresponding linear fits (solid lines).

**Figure 4 antioxidants-12-02002-f004:**
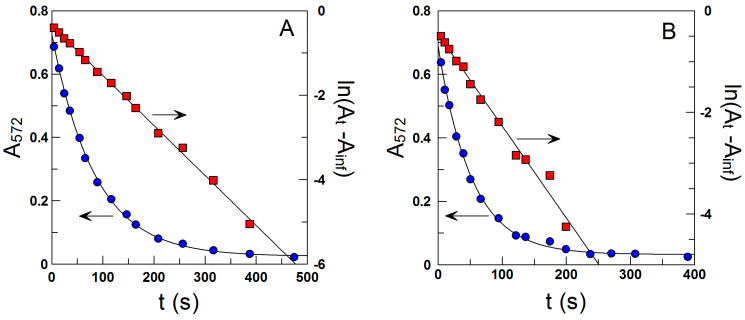
Illustrative kinetic plots showing the determination of the observed rate constant for the reaction of 16-ArN_2_^+^ with the C_1_ (**A**) and C_16_ (**B**) HT derivatives in intact 1:9 fish oil-in-water emulsions. Experimental conditions: Surfactant (Tween 80) volume fraction Φ_I_ = 0.0284), [16-ArN_2_^+^] ≈ 2.9 × 10^−4^ M, [AO_T_] = 2.5 × 10^−3^ M, T = 25 °C.

**Figure 5 antioxidants-12-02002-f005:**
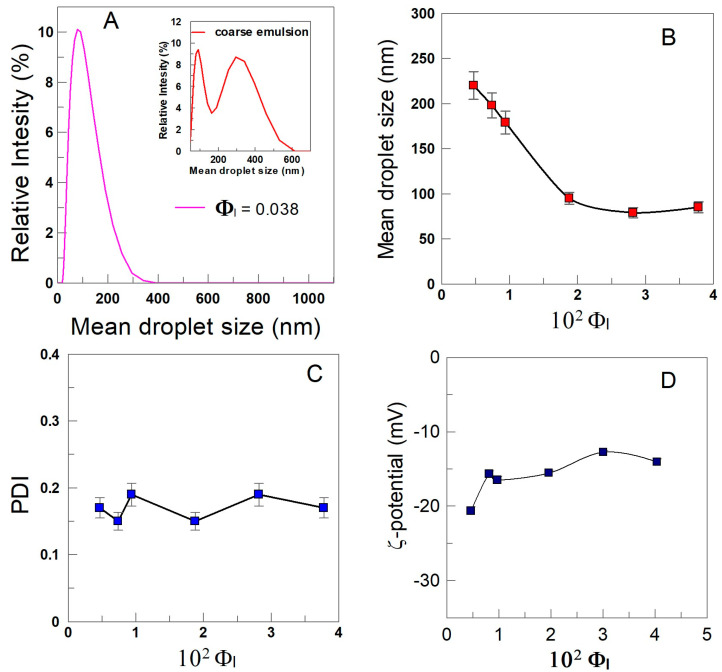
(**A**) Distribution size (D3,2) for a 1:9 (*o*/*w*) fish oil nanoemulision at Φ_I_ = 0.038 (inset: distribution size for a fish oil coarse emulsion prepared by employing the same emulsifier volume fraction). (**B**) Variation of the mean droplet size (D3,2), (**C**) polydispersity index (PDI), and (**D**) ζ-potential with the emulsifier volume fraction (Φ_I_) in 1:9 (*o*/*w*) fish oil nanoemulsions.

**Figure 6 antioxidants-12-02002-f006:**
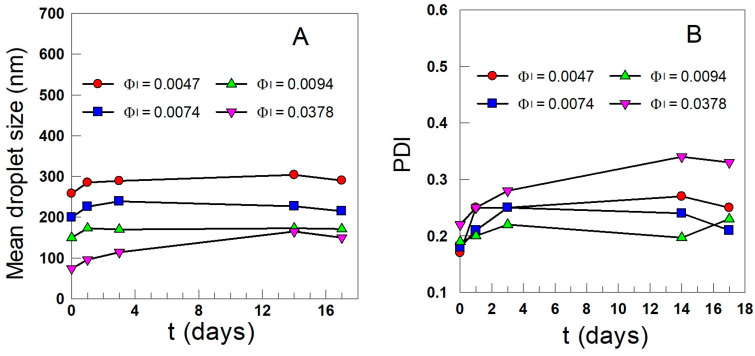
Variation of the mean droplet size (D3,2, (**A**)) and of the polydispersity index (**B**) of 1:9 (*o*/*w*) fish oil nanoemulsions stabilized with different amounts of Tween 80 with time.

**Figure 7 antioxidants-12-02002-f007:**
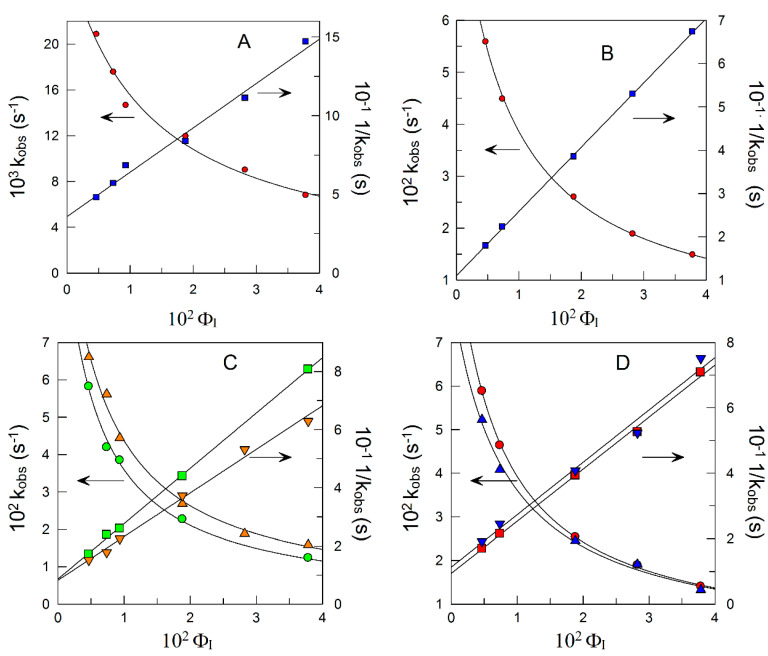
Illustrative examples of the variations of *k*_obs_ and 1/*k*_obs_ with Φ_I_ for the reaction of 16-ArN_2_^+^ with HT (**A**), C3 (**B**), C8 (●), C10 (▲) (**C**), C12 (●), and C16 (▼) (**D**) in intact 1:9 fish oil nanoemulsions as determined by the azo-dye derivatization method (see Section 2.2.5). Experimental conditions: [16-ArN_2_^+^] = 2.8 × 10^−4^ M, [AO] = 2.5 × 10^−3^ M, [NED] = 0.02 M, T = 25 °C.

**Figure 8 antioxidants-12-02002-f008:**
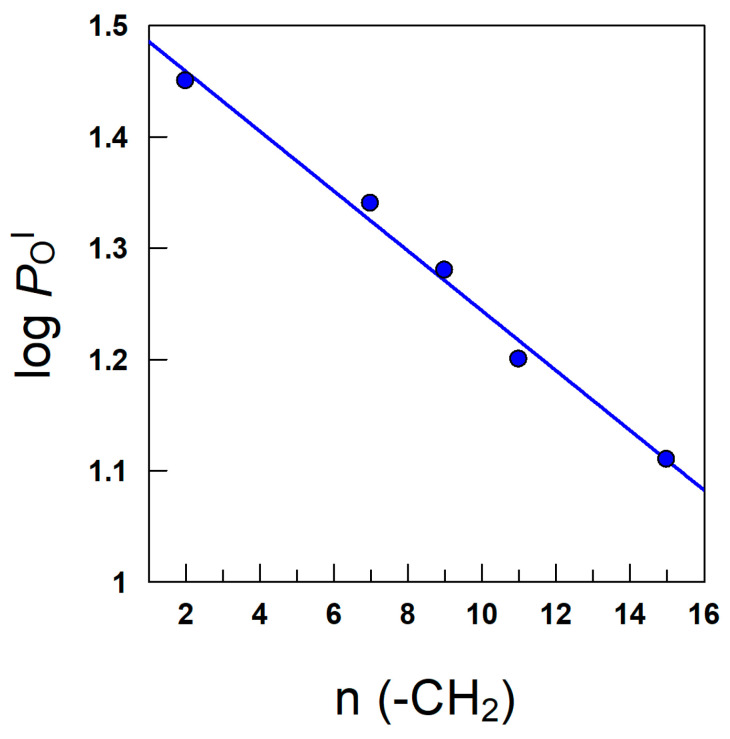
A linear relationship showing the increasing contribution of methylene groups in the HT alkyl chains to the value of the partition constant according to Equation (4).

**Figure 9 antioxidants-12-02002-f009:**
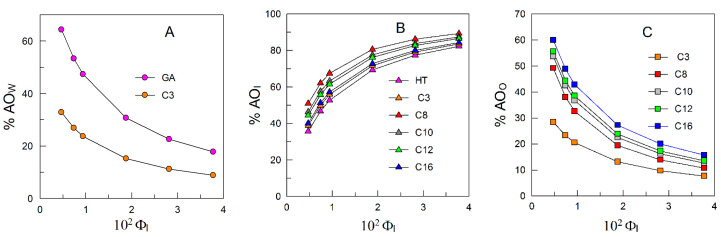
Changes in percentages of HT and HT esters as a function of Φ_I_ in the (**A**) aqueous (%AO_W_), (**B**) interfacial (%AO_I_), and (**C**) organic regions (%AO_O_). Distribution data were calculated from the partition constants in Table 3, determined at T = 25 °C.

**Figure 10 antioxidants-12-02002-f010:**
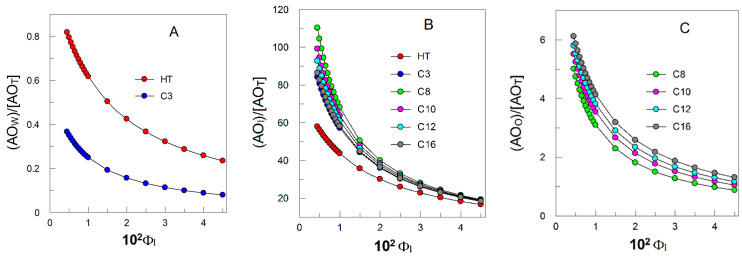
Changes in the fraction of the AOs located in the different regions ((**A**) aqueous, (**B**) interfacial, and (**C**) oil) of the 1:9 fish oil-in-water nanoemulsions with the interfacial volume fraction Φ_I_.

**Figure 11 antioxidants-12-02002-f011:**
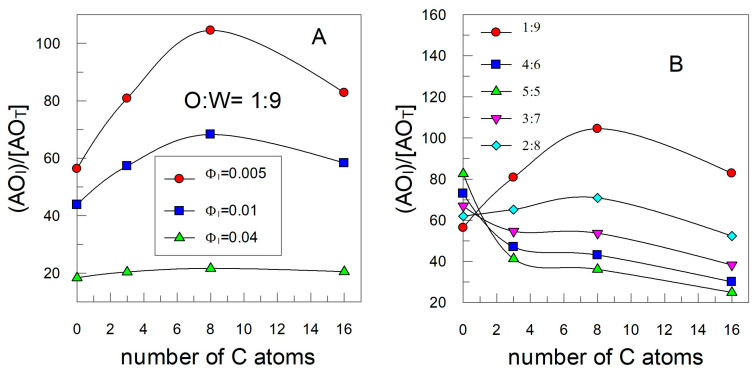
(**A**) Variation in the interfacial antioxidant fraction with the hydrophobicity of AO in fish oil-in-water nanoemulsions prepared with (**A**) three different emulsifier volume fractions (Φ_I_) and at a constant *o*/*w* ratio and (**B**) different *o*/*w* ratios and at constant Φ_I_ = 0.005.

**Figure 12 antioxidants-12-02002-f012:**
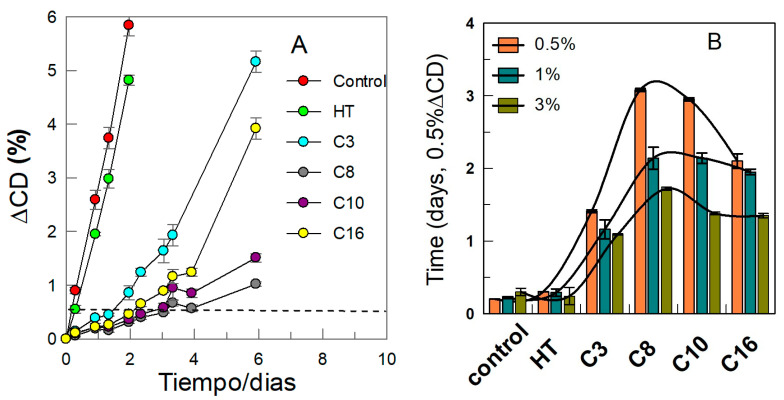
(**A**) Kinetic profiles for the variation of the conjugated dienes formed with the time in 1:9 fish oil-in-water nanoemulsions stored in the absence and in the presence of HT and its derivatives at T = 25 °C (Φ_I_ = 0.005, [AO_T_] = 0.1 mM in the total volume of the nanoemulsion). (**B**) Effect of the emulsifier volume fraction Φ_I_ on the time needed to reach 0.5% of conjugated dienes in the absence and in the presence of the different AOs in nanoemulsions prepared with Φ_I_ = 0.5%, 1%, and 3%.

**Figure 13 antioxidants-12-02002-f013:**
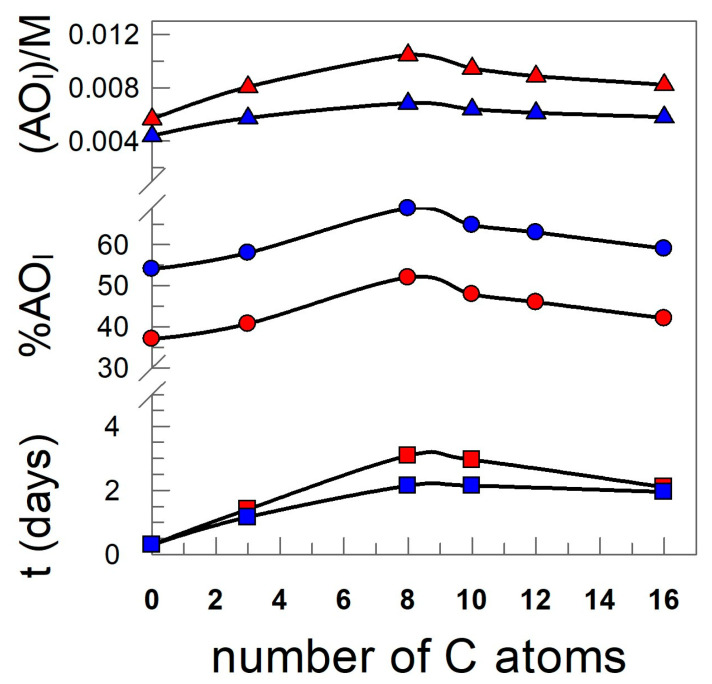
Variation of the percentage of AOs in the interfacial region (%AO_I_) of the effective interfacial concentration and antioxidant efficiency, expressed in terms of the time required to reach 0.5% CD with the hydrophobicity of the AO in fish oil-in-water nanoemulsions prepared by employing two different emulsifier volume fractions (red symbols ●, ■, ▲), Φ_I_ = 0.005 (blue symbols, ●, ■, ▲) Φ_I_ = 0.01. [AO_T_] = 0.1 mM.

**Table 1 antioxidants-12-02002-t001:** Values of EC_50_ (mol AO/mol DPPH^●^) for HT and its hydrophobic esters at different reaction times. T = 25 °C.

AO	EC_50_			
*t* (min)	5	15	30	60
HT	0.323 ± 0.005	0.287 ± 0.004	0.279 ± 0.004	0.258 ± 0.004
C_1_	0.338 ± 0.005	0.296 ± 0.003	0.288 ± 0.003	0.277 ± 0.003
C_3_	0.345 ± 0.004	0.299 ± 0.002	0.289 ± 0.005	0.280 ± 0.003
C_8_	0.295 ± 0.005	0.265 ± 0.005	0.256 ± 0.007	0.239 ± 0.003
C_10_	0.326 ± 0.005	0.292 ± 0.005	0.286 ± 0.007	0.280 ± 0.004
C_12_	0.345 ± 0.005	0.319 ± 0.004	0.309 ± 0.011	0.294 ± 0.006
C_16_	0.331 ± 0.011	0.305 ± 0.006	0.302 ± 0.011	0.298 ± 0.007

**Table 2 antioxidants-12-02002-t002:** Percentages of HT and of some hydrophobic esters present in the aqueous (%AO_W_) and oil (%AO_O_) phases of binary 1:9 (v:v) mixtures and average (four runs) values of the partition constant *P*_W_^O^.

AO	%AO_W_	%AO_O_	*P* _W_ ^O^
HT	96.99	3.01	0.3 ± 0.1
C2	70.65	29.35	3.7 ± 0.1
C3	53.63	46.37	7.8 ± 0.4
C6	5.31	94.69	162 ± 19

**Table 3 antioxidants-12-02002-t003:** Values of the interfacial rate constant for the reaction between 16ArN_2_^+^ and AOs, *k*_I_, and of the partition constants of HT and its derivatives determined in intact fish oil-in-water nanoemulsions (T = 25 °C).

AO	10^2^ *k*_I_ (M^−1^ s^−1^)	*P* _O_ ^I^	*P* _W_ ^I^
HT	14.2 ± 0.1	…	71 ± 12
C3	27.7 ± 4.7	28 ± 3	229 ± 40
C8	21.0 ± 0.3	22 ± 2	…
C10	26.6 ± 1.1	19 ± 4	…
C12	25.3 ± 0.7	16 ± 2	…
C16	25.0 ± 0.8	13 ± 2	…

## Data Availability

Raw data can be obtained upon request at any masthead.

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
