# Peer review of "Exploring the Use of Hydroxytyrosol and Some of Its Esters in Food-Grade Nanoemulsions: Establishing Connection between Structure and Efficiency"

_antioxidants, 2023, doi:10.3390/antiox12112002_

Round 1

Reviewer 1 Report

Comments and Suggestions for Authors

The manuscript titled "Investigating Hydroxytyrosol and Its Esters in Fish Oil-in-Water Nanoemulsions: Establishing Structure-Efficiency Relationships" presents an intriguing exploration of the application of hydrophobic esters and their effectiveness when incorporated into oil-in-water nanoemulsions. This research aligns well with the journal Antioxidants and its Special Issue, "Advanced Strategies for the Oxidative Stabilization of Wet and Dry Emulsions." While the concept of hydrophobic esters of hydroxytyrosol is not entirely novel, their correlation with efficiency in an oil-in-water system, particularly within the realm of food colloids, piques significant interest. In consideration of publication, I recommend a Major Revision to address the following key points:

Major Points

1. Title refining could be useful to make it more concise and reflective of the research's core focus, for example, "Exploring Hydroxytyrosol and Its Esters in Food-Grade Nanoemulsions: Establishing a Connection between Structure and Efficiency."

 2. The manuscript would benefit from language editing to eliminate redundancy and enhance clarity. Some sentences contain repetitive concepts that could be consolidated for improved readability. For example in lines 37 & 42 are repeating the same concept two times very close in the text.

 3. Ensure the references are correctly ordered within the text. For example Ref.14 appears in the text before Ref. 13. 

4. Consider the necessity of Figure 1. It doesn't substantially contribute to readers' understanding, and replacing it with a "graphical abstract" might be more beneficial, particularly for readers familiar with hydroxytyrosol and related antioxidants. Otherwise, please remove it totally.

 5. The introduction should be revised to clearly state the primary research goals. Emphasize the main objective of the study at the end of the section (last paragraph) and then comment on the overall goal. 

 6.  Provide precise details regarding material purity. Specify equipment, temperature, and other conditions employed during Dynamic Light Scattering measurements. Move results-related content, such as nanoemulsion diameter and kinetic plots, to the Results section.

 7. Include the polydispersity index and z-potential measurements for the nanoemulsions before and after compound incorporation in all cases. Additionally, provide Dynamic Light Scattering measurements to assess the 24-hour "stability" of the systems, as visual inspection alone is not be adequate.

 8. Analyze the impact of the emulsification process on the antioxidant activity of the compounds. Specify the temperature of the procedure and present antioxidant activity values before and after homogenization. Each component could behave differently.  

9. Clearly state the stability of the system and present experimental data detailing how the compounds efficiency change over time. If the system exhibits stability over several days, demonstrate the degradation profiles of the compounds. In the case of instability, provide a rationale for using this specific system.

Minor Points

1. The abstract could be made more concise by eliminating the introductory sentence and focusing on the primary topic and conclusion.

2. Add "lipid oxidation" to the list of keywords, and replace "nanoemulsions" with "food-grade nano-emulsions."

3. Include a brief literature review in the introduction to provide readers with context on the relevant research regarding hydroxytyrosol and esters in colloidal systems. Suggested literature: (1) Lucas, Ricardo, et al. "Surface-active properties of lipophilic antioxidants tyrosol and hydroxytyrosol fatty acid esters: A potential explanation for the nonlinear hypothesis of the antioxidant activity in oil-in-water emulsions." Journal of agricultural and food chemistry 58.13 (2010): 8021-8026. (2) Mitsou, Evgenia, et al. "Hydroxytyrosol encapsulated in biocompatible water-in-oil microemulsions: How the structure affects in vitro absorption." Colloids and Surfaces B: Biointerfaces 184 (2019): 110482. (3) Cuomo, Francesca, et al. "Progress in colloid delivery systems for protection and delivery of phenolic bioactive compounds: two study cases—hydroxytyrosol and curcumin." Molecules 27.3 (2022): 921.

Comments on the Quality of English Language

Minor English editing 

Author Response

REVIEWER 1

The manuscript titled "Investigating Hydroxytyrosol and Its Esters in Fish Oil-in-Water Nanoemulsions: Establishing Structure-Efficiency Relationships" presents an intriguing exploration of the application of hydrophobic esters and their effectiveness when incorporated into oil-in-water nanoemulsions. This research aligns well with the journal Antioxidants and its Special Issue, "Advanced Strategies for the Oxidative Stabilization of Wet and Dry Emulsions." While the concept of hydrophobic esters of hydroxytyrosol is not entirely novel, their correlation with efficiency in an oil-in-water system, particularly within the realm of food colloids, piques significant interest. In consideration of publication, I recommend a Major Revision to address the following key points:

Many thanks for your comments, which certainly helped us to improve the clarity of the manuscript. We considered all your suggestions, and details are as follows.

Major Points

  1. Title refining could be useful to make it more concise and reflective of the research's core focus, for example, "Exploring Hydroxytyrosol and Its Esters in Food-Grade Nanoemulsions: Establishing a Connection between Structure and Efficiency."

We like the suggestion made by the reviewer and consequently we changed the title as suggested.

  1. The manuscript would benefit from language editing to eliminate redundancy and enhance clarity. Some sentences contain repetitive concepts that could be consolidated for improved readability. For example in lines 37 & 42 are repeating the same concept two times very close in the text.

We have gone again throught out the text and revised accordingly.

  1. Ensure the references are correctly ordered within the text. For example Ref.14 appears in the text before Ref. 13. 

References within the text had been revised accordingly and are now in proper order.

  1. Consider the necessity of Figure 1. It doesn't substantially contribute to readers' understanding, and replacing it with a "graphical abstract" might be more beneficial, particularly for readers familiar with hydroxytyrosol and related antioxidants. Otherwise, please remove it totally.

We removed figure 1 from the text and placed it as graphical abstracts following your recommendations.

  1. The introduction should be revised to clearly state the primary research goals. Emphasize the main objective of the study at the end of the section (last paragraph) and then comment on the overall goal. 

We revised thoroughly the introduction emphasizing at the end of the section the objectives of the paper.

  1. Provide precise details regarding material purity.

Details about RMN spectra of HT derivatives are given in the Supporting Information (section S1)

Specify equipment, temperature, and other conditions employed during Dynamic Light Scattering measurements.

Done.

Move results-related content, such as nanoemulsion diameter and kinetic plots, to the Results section.

The reviewer must have in mind that the main aim of the paper is not to report on the physical properties of nanoemulsions. This has already been investigated in previous works (and properly referenced in the text). Thus, we consider those results are not so relevant to place them in the results section, which is focused in those much closer to the main aim of the manuscript. Thus, we think that figures 4 and 5 should remain in the methododlogy section “Methods” section for a better flow of the manuscript.  However, results related with the characterization and the physical stability of nanoemulsions were added to Results and Discussion section for clarity.

  1. Include the polydispersity index and z-potential measurements for the nanoemulsions before and after compound incorporation in all cases. Additionally, provide Dynamic Light Scattering measurements to assess the 24-hour "stability" of the systems, as visual inspection alone is not be adequate.

Done. Following your suggestions, we have included a brief description about the characterization and the physical stability of nanoemulsions in “Results and Discussion section”.

  1. Analyze the impact of the emulsification process on the antioxidant activity of the compounds. Specify the temperature of the procedure and present antioxidant activity values before and after homogenization. Each component could behave differently.  

Thanks for your comment but, we are afraid we do not get your point. The aim of the manuscript is to investigate the efficiency of a series of HT esters in the naoemulsions. Without the emulsification processs, the system is totally different and, in our opinion, results in binary oil-water systems cannot be compared with those in emulsions because of the critical role of the interfacial region. Thus, we cannot compare results before and after homogenization.

  1. Clearly state the stability of the system and present experimental data detailing how the compounds efficiency change over time. If the system exhibits stability over several days, demonstrate the degradation profiles of the compounds. In the case of instability, provide a rationale for using this specific system.

Previous experiments focused on the physical stability of the nanoemulsions showed are the prepared nanoemulsions are rather stable with ageing for more than 2 weeks, a much longer time than that required to complete the kinetic experiments about the chemical stability of nanoemulsions in the absence and in the presence of antioxidants. Besides, nanoemulsions were placed in a thermostated orbital shaker, maintaining agitation during overall chemical stability experiment to further ensure their physical stability.

We agree with the reviewer that antioxidant efficiency differs significantly when switching from an oil/water binary system to an emulsified system but the uniformity of the induction period for different antioxidants indicates minimal effects of phase separation on the kinetics of oxidation because, otherwise, the values should have much more scatter in their variation with time.

Minor Points

  1. The abstract could be made more concise by eliminating the introductory sentence and focusing on the primary topic and conclusion.

Done.

  1. Add "lipid oxidation" to the list of keywords, and replace "nanoemulsions" with "food-grade nano-emulsions."

Done.

  1. Include a brief literature review in the introduction to provide readers with context on the relevant research regarding hydroxytyrosol and esters in colloidal systems. Suggested literature: (1) Lucas, Ricardo, et al. "Surface-active properties of lipophilic antioxidants tyrosol and hydroxytyrosol fatty acid esters: A potential explanation for the nonlinear hypothesis of the antioxidant activity in oil-in-water emulsions." Journal of agricultural and food chemistry 58.13 (2010): 8021-8026. (2) Mitsou, Evgenia, et al. "Hydroxytyrosol encapsulated in biocompatible water-in-oil microemulsions: How the structure affects in vitro absorption." Colloids and Surfaces B: Biointerfaces 184 (2019): 110482. (3) Cuomo, Francesca, et al. "Progress in colloid delivery systems for protection and delivery of phenolic bioactive compounds: two study cases—hydroxytyrosol and curcumin." Molecules 27.3 (2022): 921.

Done.

Reviewer 2 Report

Comments and Suggestions for Authors

1. Figures 1, 2, and 3 should be the result of citing other literature, so there should be a citation in the figure title.

2. Line 292-294, if it is a constant, it is recommended to write down the actual value used in this research.

3. This manuscript is incomplete. There are results in the manuscript, but there is a lack of discussion.

4. The  first  occurrence  of  AO  is  not  expressed  in  full  letters.  Please  write  the  full  letters  of  AO.

5. In  page  2  line  44,  the  word  ’’benezene-1,2-diol’’  should  be  corrected  to  ‘’benzene-1,2-diol’’

6. following  this  modification  process,  has  any  experimentation  been  conducted  to  establish  its  suitability  and  safety  as  a  food  additive?

7. This  study  aims  to  expand  the  use  of  HT  in  cardiovascular  disease  and  utilize  olive  oil  industry  by-products,  which  impact  economic  sustainability.  But  the  importance  of  this  issue  was  not  explicitly  mentioned  or  emphasized  in  the  introductory  section.

8. The  information  on  fish  oil  and  nanoemulsions  should  be  provided  more  in  the  introduction.

9. The  full  term  of  AO,  AOs,  AOt,  AOi,  and  all  other  abbreviations  should  be  given  at  its  first  mention.

10. Section  3.3,  Please  clarify  what  the  "control"  sample  in  this  study  is.

11. In  conclusion,  it  is  recommended  to  write  a  response  to  the  hypothesis  and  objective  statement,  emphasizing  its  potential,  the  opportunity  for  industrial  expansion,  and  economic  sustainability.

12. Line  499-502:  Please  elaborate  more  about  the  reason  of  this  phenomenon.

13. Why  this  research  only  uses  some  of  esters  from  HT  (C1,  C3,  C8,  C10,  C12,  C16)  ?

14. In  line  496,  please  explain  the  "cut-off"  effect  depicted  in  Figure  11B,  expounding  upon  its  representative  significance.  Also,  could  you  explain  the  representative  implications  associated  with  the  legend  items  denoted  as  0.5%,  1%,  and  3%  within  Figure  11B?

15. The  prepared  nanoemulsion  is  recommended  for  stability  and  dispersibility  tests  of  the  particles.

16. To  provide  a  brief  overview  of  the  mechanisms  underlying  the  various  functionalities  of  Hydroxytyrosol  (HT),  such  as  its  applications  in  antioxidation,  antimicrobial  effects,  and  anti-inflammatory  properties,  presented  in  the  form  of  a  literature  review.

17. Please  explain  why  Figure  11B  uses  0.5%  conjugated  dienes(0.5%ΔCD)  as  the  standard  ?

18. In  figure  8C  and  9C,  why  HT  is  not  included  in  the  experiment  ?

19. In  Figure  4,  the  unit  of  DPPH  clearance  rate  is  %,  so  the  y-axis  values  should  be  40%,  60%,  80%,  and  100%.

20. In  Figure  4,  please  add  the  error  bar;  In  Table  1,  there  should  be  indicated  significant  differences  with  hydrophobic  esters  at  different  reaction  times.

21. Comparing  proposed  additives  with  the  currently  prevalent  agents  utilized  for  the  preservation  of  fish  oil.

22. The  conclusion  should  be  its  own  chapter.

23. Please  add  additional  explanation  why  it  cannot  be  used  by  heating,  or  attach  a  reference  at  Line  70. 

Author Response

REVIEWER 2

Comments and Suggestions for Authors

Many thanks for your time and the efforts made to improve the manuscript, we appreciate them. We considered most of your suggestions and you can find details below

  1. Figures 1, 2, and 3 should be the result of citing other literature, so there should be a citation in the figure title.

Figures 1-3 are original figures of this manuscript.

  1. Line 292-294, if it is a constant, it is recommended to write down the actual value used in this research.

Done

  1. This manuscript is incomplete. There are results in the manuscript, but there is a lack of discussion.

Sorry, but we do not get the point. We agree that there is not discussion section as such, but results are discussed, compared to those in the literature (when available), and place in context just after showing them in the text.

Furthermore, we followed the suggestions of  reviewer 1 and included some discussion about characterization and physical stability of the prepared nanoemulsions.

  1. The  first  occurrence  of  AO  is  not  expressed  in  full  letters.  Please  write  the  full  letters  of  AO.

Done

  1. In  page  2  line  44,  the  word  ’’benezene-1,2-diol’’  should  be  corrected  to  ‘’benzene-1,2-diol’’

Done

  1. following  this  modification  process,  has  any  experimentation  been  conducted  to 

 establish  its  suitability  and  safety  as  a  food  additive?

No, not at this stage. The focus of the present paper is to investigate the effects of hydrophonbicity on the efficiency of antioxidants. Further experiments need to be done to test them as suitable food addivives if considered appropriate.

  1. This  study  aims  to  expand  the  use  of  HT  in  cardiovascular  disease  and  utilize  olive  oil  industry  by-products,  which  impact  economic  sustainability.  But  the  importance  of  this  issue  was  not  explicitly  mentioned  or  emphasized  in  the  introductory  section.

We think the reviewer missed the point. We mention the possibility of using HT and some of its derivatives in human health in the introduction, but it is not the point of this manuscript and so we think it is not necessary further discussion on the topic.

  1. The  information  on  fish  oil  and  nanoemulsions  should  be  provided  more  in  the  introduction.

We do not understand what the reviewer means and why more information on nanoemulsion preparation should be given in the introduction. Information related to the preparation of the nanoemulsions is already given in in the methodological section. No need of further discussion in the introduction. This has the advantage of focusing on the topic clearly and avoid missinterpreations as the preparation of the nanoemulsions is not the aim of the manuscript.

  1. The  full  term  of  AO,  AOs,  AOt,  AOi,  and  all  other  abbreviations  should  be  given  at  its  first  mention.

Done.

  1. Section  3.3,  Please  clarify  what  the  "control"  sample  in  this  study  is.

Done.

  1. In  conclusion,  it  is  recommended  to  write  a  response  to  the  hypothesis  and  objective  statement,  emphasizing  its  potential,  the  opportunity  for  industrial  expansion,  and  economic  sustainability.

Done.

  1. Line  499-502:  Please  elaborate  more  about  the  reason  of  this  phenomenon.

Done. We added a paragraph in page 16-17 of the revised manuscript to clarify this point.

  1. Why  this  research  only  uses  some  of  esters  from  HT  (C1,  C3,  C8,  C10,  C12,  C16)  ?

We prepared some HT derivatives, those that could be easily synthetized and, at the same time, cover a wide range of hydrophobicities.

  1. In  line  496,  please  explain  the  "cut-off"  effect  depicted  in  Figure  11B,  expounding  upon  its  representative  significance. 

Thanks for drawing our attention to this point. We have already explained the “cut-off” in previous reports on the basis of the differential concentration of the antioxidants in the interfacial region, see references in the text. So, we think it is not necessary to repeat it again.

Also,  could  you  explain  the  representative  implications  associated  with  the  legend  items  denoted  as  0.5%,  1%,  and  3%  within  Figure  11B?

Done . This point was clarified.

  1. The  prepared  nanoemulsion  is  recommended  for  stability  and  dispersibility  tests  of  the  particles.

Following your suggestions and those of reviewer 1, we have included a brief description about the characterization and the physical stability of nanoemulsions in “Results and Discussion section”.

  1. To  provide  a  brief  overview  of  the  mechanisms  underlying  the  various  functionalities  of  Hydroxytyrosol  (HT),  such  as  its  applications  in  antioxidation,  antimicrobial  effects,  and  anti-inflammatory  properties,  presented  in  the  form  of  a  literature  review.

We updated the references to include some related to the topics you mention. However, we did not expand the text on them because they are not relevant for the purposes of the manuscript-

  1. Please  explain  why  Figure  11B  uses  0.5%  conjugated  dienes(0.5%ΔCD)  as  the  standard  ?

It is mentioned in the text that the method was adopted because a world-wide interlaboratory study concluded that the method is trustable and can be employed to monitor the formation of primary oxidation products. The experiments are properly referenced in the text.

  1. In  figure  8C  and  9C,  why  HT  is  not  included  in  the  experiment  ?

It is not included HT in both figures due to HT is a hydrophilic antioxidant and thus, its percentage in the oil region is ~ 0.

  1. In  Figure  4,  the  unit  of  DPPH  clearance  rate  is  %,  so  the  y-axis  values  should  be  40%,  60%,  80%,  and  100%.

The reviewer is right. Thank you for your sharp observation and drawing our attention to the point.

  1. In  Figure  4,  please  add  the  error  bar;  In  Table  1,  there  should  be  indicated  significant  differences  with  hydrophobic  esters  at  different  reaction  times.

Figure 4 shows typical representations to obtain EC50 values for this reason error bar are not included here.  At the initial reaction time, the differences between AOs are not significantly.

  1. Comparing  proposed  additives  with  the  currently  prevalent  agents  utilized  for  the  preservation  of  fish  oil.

We do not get the point. We do not use preservatives. Precisely, we investigate the oxidation status of the nanoemulsions when loaded with different antioxidants.

  1. The  conclusion  should  be  its  own  chapter.

Done

  1. Please  add  additional  explanation  why  it  cannot  be  used  by  heating,  or  attach  a  reference  at  Line  70.

Reference was included.

Round 2

Reviewer 1 Report

Comments and Suggestions for Authors

The manuscript has been substantially improved and it could be published in the present form. 

Author Response

Reviewer 1

Many thanks for your time and efforts to improve the manuscript. Since you did not suggest any other comment, no further actions were taken.

Reviewer 2 Report

Comments and Suggestions for Authors

The author revised based on my suggestions, and I agree with the improvements the author made to focus the manuscript. However, the author uses nanoemulsion as the title, but lacks the current status of fish oil in nanoemulsions and related research (problems and directions for improvement should be explained clearly), which I think weakens the necessity of this research. Although some studies in the manuscript supports that nanoemulsions can reduce oxidative deterioration. After all, most fish oil products are currently on the market in capsule form. Are nanoemulsions a suitable/common fish oil carrier?

This is also the reason why I hoped that the author would add information about nanoemulsions when I reviewed the manuscript for the first time.

I agree that this is a good study, from the experimental design and writing logic. Therefore, I hope that the author can further explain why nanoemulsion is chosen as an experimental platform to highlight the advantages of HT-stabilized fish oil to complete the value of this research discovery. 

Author Response

Reviewer 2

The author revised based on my suggestions, and I agree with the improvements the author made to focus the manuscript. However, the author uses nanoemulsion as the title, but lacks the current status of fish oil in nanoemulsions and related research (problems and directions for improvement should be explained clearly), which I think weakens the necessity of this research. Although some studies in the manuscript supports that nanoemulsions can reduce oxidative deterioration. After all, most fish oil products are currently on the market in capsule form. Are nanoemulsions a suitable/common fish oil carrier?

This is also the reason why I hoped that the author would add information about nanoemulsions when I reviewed the manuscript for the first time.

I agree that this is a good study, from the experimental design and writing logic. Therefore, I hope that the author can further explain why nanoemulsion is chosen as an experimental platform to highlight the advantages of HT-stabilized fish oil to complete the value of this research discovery. 

R.

Thanks for your suggestion. Following it, we added the following paragraph before the last paragraph of the introduction section:

We have chosen nanoemulsions as experimental platform in our studies because Nanoemulsions constitute advanced mode of drug delivery system has been developed to overcome the major drawbacks associated with conventional drug delivery[1-7]. Because the food industry is now introducing omega-3 FA to prepare various kinds of functional foods in attempting to provide health benefits over and above their basic nutritional aspects, there are considerable challenges in incorporating both antioxidants and omega-3 FAs into many types of functional food products due to their low water-solubility, poor chemical stability, and variable bioavailability[8]. Consequently, there has been growing interest in the development of appropriate delivery systems to encapsulate, protect, and release them[8,9]. 

Nanoemulsions offer a promising way to incorporate omega-3 fatty acids into liquid food systems like beverages, dressing, sauces, and dips. The composition and fabrication of nanoemulsions can be optimized to increase the chemical and physical stability of oil droplets, as well as to increase the bioavailability of omega-3 fatty acids. Delivery systems such as nanoemulsions could be used for a number of purposes: controlling lipid bioavailability; targeting the delivery of bioactive components within the gastrointestinal tract; and designing food matrices that delay lipid digestion and induce and functional lipophilic constituents within the food and pharmaceutical industries. Nanoemulsion drug delivery systems can be, thus, envisaged as advanced modes for delivering and improving the bioavailability of hydrophobic drugs and the drug which have high first pass metabolism. The nanoemulsion can be prepared by both high energy and low energy methods, providing an excellent working system for optimal drug delivery for existing and newly developed antioxidants and antimicrobials, enhancing drug bioavailability, enabling site-specific drug targeting, and overcoming current limitations of drug formulations such as short elimination half-lives, poor drug solubility, and undesirable side effects[10-12]. Choosing nanoemulsions as working platform has, at the same time, the advantage of attempting to fill the need for edible delivery systems to encapsulate, protect and release bioactive[1,2,9,13].

References included in the revised text

  1. Chatzidaki, M.D.; Mitsou, E.; Yaghmur, A.; Xenakis, A.; Papadimitriou, V. Formulation and characterization of food-grade microemulsions as carriers of natural phenolic antioxidants. Colloids and Surfaces A: Physicochemical and Engineering Aspects 2015, 483, 130-136, doi:https://doi.org/10.1016/j.colsurfa.2015.03.060.
  2. Chatzidaki, M.D.; Arik, N.; Monteil, J.; Papadimitriou, V.; Leal-Calderon, F.; Xenakis, A. Microemulsion versus emulsion as effective carrier of hydroxytyrosol. Colloids and Surfaces B: Biointerfaces 2016, 137, 146-151, doi:https://doi.org/10.1016/j.colsurfb.2015.04.053.
  3. Galani, E.; Galatis, D.; Tzoka, K.; Papadimitriou, V.; Sotiroudis, T.G.; Bonos, A.; Xenakis, A.; Chatzidaki, M.D. Natural Antioxidant-Loaded Nanoemulsions for Sun Protection Enhancement. Cosmetics 2023, 10, 102.
  4. Colucci, G.; Santamaria-Echart, A.; Silva, S.C.; Fernandes, I.P.M.; Sipoli, C.C.; Barreiro, M.F. Development of Water-in-Oil Emulsions as Delivery Vehicles and Testing with a Natural Antimicrobial Extract. Molecules 2020, 25, 2105.
  5. Bhatti, H.S.; Khalid, N.; Uemura, K.; Nakajima, M.; Kobayashi, I. Formulation and characterization of food grade water-in-oil emulsions encapsulating mixture of essential amino acids. European Journal of Lipid Science and Technology 2017, 119, 1600202, doi:https://doi.org/10.1002/ejlt.201600202.
  6. Ribeiro, T.B.; Bonifácio-Lopes, T.; Morais, P.; Miranda, A.; Nunes, J.; Vicente, A.A.; Pintado, M. Incorporation of olive pomace ingredients into yoghurts as a source of fibre and hydroxytyrosol: Antioxidant activity and stability throughout gastrointestinal digestion. Journal of Food Engineering 2021, 297, 110476, doi:https://doi.org/10.1016/j.jfoodeng.2021.110476.
  7. Pintado, T.; Muñoz-González, I.; Salvador, M.; Ruiz-Capillas, C.; Herrero, A.M. Phenolic compounds in emulsion gel-based delivery systems applied as animal fat replacers in frankfurters: Physico-chemical, structural and microbiological approach. Food Chem. 2021, 340, 128095, doi:https://doi.org/10.1016/j.foodchem.2020.128095.
  8. Walker, R.; Decker, E.A.; McClements, D.J. Development of food-grade nanoemulsions and emulsions for delivery of omega-3 fatty acids: opportunities and obstacles in the food industry. Food & Function 2015, 6, 41-54, doi:10.1039/C4FO00723A.
  9. Demisli, S.; Chatzidaki, M.D.; Xenakis, A.; Papadimitriou, V. Recent progress on nano-carriers fabrication for food applications with special reference to olive oil-based systems. Current Opinion in Food Science 2022, 43, 146-154, doi:https://doi.org/10.1016/j.cofs.2021.11.012.
  10. Garcia, C.R.; Malik, M.H.; Biswas, S.; Tam, V.H.; Rumbaugh, K.P.; Li, W.; Liu, X. Nanoemulsion delivery systems for enhanced efficacy of antimicrobials and essential oils. Biomaterials Science 2022, 10, 633-653, doi:10.1039/D1BM01537K.
  11. Jaiswal, M.; Dudhe, R.; Sharma, P.K. Nanoemulsion: an advanced mode of drug delivery system. 3 Biotech 2015, 5, 123-127, doi:10.1007/s13205-014-0214-0.
  12. Kumar, M.; Bishnoi, R.S.; Shukla, A.K.; Jain, C.P. Techniques for Formulation of Nanoemulsion Drug Delivery System: A Review. Prev Nutr Food Sci 2019, 24, 225-234, doi:10.3746/pnf.2019.24.3.225.
  13. Wilson, R.J.; Li, Y.; Yang, G.; Zhao, C.-X. Nanoemulsions for drug delivery. Particuology 2022, 64, 85-97, doi:https://doi.org/10.1016/j.partic.2021.05.009.